# OpenReview forum: "SecureAgentBench: Benchmarking Secure Code Generation under Realistic Vulnerability Scenarios"
_ICLR.cc/2026/Conference — ICLR 2026 Conference Withdrawn Submission_

### Official Review · Reviewer_hdwE · 2025-10-21

**Soundness:** 2
**Presentation:** 2
**Contribution:** 3
**Rating:** 4
**Confidence:** 5

**Summary:**

This paper introduces a repository-level coding agent benchmark aimed at measuring both functionality and memory safety of the agent-generated code. The benchmark is constructed from repositories where OSS-Fuzz has discovered vulnerabilities before, but have been fixed since. After several rounds of heuristic and partially manual post-processing and filtering, 105 coding tasks are selected for the benchmark. Each task consists of a description of the intended modification on the codebase, a functional test suite sourced from the original repository, and PoC programs for reproducing the vulnerability found by OSS-Fuzz. Then agents are then instructed to execute the coding instruction and are measured on the implementation's functionality and security. The evaluation across three agent scaffolds and three LLMs concludes that even in the best examined setup, barely over $15$% of the generated code is secure and correct.

**Strengths:**

- Measuring the security of coding agent-generated code on repository-level tasks is important, and has been overlooked by most previous code security benchmarks that focus mostly on function-level or standalone, small, application code.
- Focusing on the code repository state before the vulnerability has been *introduced* in the repository and formulating the same task as the one that introduced the vulnerability is a neat approach.
- Combining dynamic and static analysis follows good practice.

**Weaknesses:**

- The scope is somewhat limited: i) only memory-safety bugs; ii) only C/C++; and iii) only 41 historic repositories. While this in itself is mostly fine, I would prefer if the paper was explicit about this. Currently, the framing sounds like more general code security evaluation. This should also be highlighted in comparison with prior benchmarks that often measure secure coding performance across different programming languages and types of vulnerabilities.
- To point iii) above; there is a certain potential concern of data contamination, which should be discussed/treated with care.
- From the statistics in Table 2, it is clear that the benchmark focuses on small amount of generated code in large, complex contexts. While this is akin to other agentic coding benchmarks (e.g., SWE-Bench) this should be explicitly mentioned in the comparisons with prior secure coding benchmarks (e.g., BaxBench requires the models to generate up to hundreds of lines of code across several files from scratch).
- I find the evaluation section, given that this is a benchmark paper, rather limited, mostly due to the lack of actual state-of-the-art systems being evaluated. Giving weight to my concerns are some counterintuitive findings, e.g., figure 5, showing that cheaper models are better. This suggests to me that the benchmark might be noisy for these lower-performant models,  and results on stronger agents+models would be more insightful (e.g., Codex + GPT5, Claude Code + Claude Sonnet 4.5). Additionally, I miss two critical further examinations of the SAST results: i) the ratio of correct but suspicious programs is rather high, but is not examined further; ii) it seems to me that the SAST result post-patch are not baselined by SAST results pre-patch and SAST results post-goldpatch. Both of these experiments would give us a clearer view of the actual ratio of new vulnerabilities introduced by the agents. Finally, I am missing a more detailed discussion on the rather surprising stark differences in the overall performance and failure patterns of the agent scaffoldings. It would be nice to link the observed performances to design choices in the agents.
- **Minor**: Presentation. I find some of the tables and figures hard/strenuous to read. Table5: uncomfortable to match the middle column to the left column---I suggest a clearer horizontal divide between the CWEs. Tables 2 and 3 are next to eachother but misaligned, creating a poor visual effect, same for Table 5 and Figure 6. Figure 4 is overloaded---I suggest focusing in the main experiment on less but higher-level metrics (e.g., C&S, CS, and a larger category of incorrect), and then have a subsequent analysis of the errors. It makes for a clearer presentation. Table 4 is dense and hard to follow the column alignments. Figure 5 violates the page formatting, I believe this is an honest error, but I advise the authors to fix it at the next possible chance. Finally, the paper has minor typos and grammatical errors that could be quickly fixed by running a spell-checker or other writing aids.

**Questions:**

Additionally to my points in "Weaknesses" above, I have the following questions:
- Why is BaxBench excluded from the comparison in Table 1? How does it compare?
- It seems that SecRepoBench already contains most of the main assets needed to construct SecureAgentBench. Why was a new data collection pipeline necessary? Could you not have just extended SecRepoBench (and potentially other prior works)?
- While the at many stages of the benchmark construction pipeline manual intervention is needed, the task descriptions are LLM-generated. Why so? It seems that this would have been a smaller manual effort then for instance verifying and setting up the functional testing suites.
- The data collection pipeline is rather lossy. Starting from almost 5k potential cases, at the end only 105 remain. What do you think, where could you improve your pipeline to decrease this loss? Could you automate more steps to have potentially a constantly evolving benchmark, reducing, e.g., contamination risks?
- For the results in Figure 7, this seems rather counterintuitive. What happens if you increase the budget constraints? What happens if you try better models/scaffolds? Are the 16 C&S cases the same or different with and without the prompt?

---

> ### Author Response · Authors · 2025-11-22
> **Official Response of Submission460 by Authors (1/N)**
>
> Many thanks for your detailed feedback and constructive suggestions. We are going to clarity the following questions:
>
> **Q1**
> > The scope is somewhat limited: i) only memory-safety bugs; ii) only C/C++; and iii) only 41 historic repositories. While this in itself is mostly fine, I would prefer if the paper was explicit about this. Currently, the framing sounds like more general code security evaluation. This should also be highlighted in comparison with prior benchmarks that often measure secure coding performance across different programming languages and types of vulnerabilities.
>
> **A1**
>
> We agree that clearly defining the scope is essential for managing reader expectations. In the revised manuscript, we will explicitly state that our work specifically focuses on memory-safety vulnerabilities within 41 C/C++ projects. Furthermore, we will incorporate additional discussion comparing our benchmark with prior works in terms of programming languages and vulnerability types, and outline future directions to position our work more clearly and fairly.
>
> While acknowledging this specific scope, we would like to highlight its significance and the rationale behind our choices:
>
> 1. C/C++ importance. C/C++ remains the dominant language for system software, and memory safety is its most critical security challenge. Industry reports from Microsoft [1] and The Chromium Projects [2] consistently show that approximately 70% of their security bugs are memory-safety issues. White House reports [3] and academic research [4] further underscore this importance.
>
> 2. Quality over quantity. Although our benchmark contains a limited number of repositories, establishing a reproducible vulnerability environment for real-world C/C++ projects is significantly more challenging than related works focusing on function completion. We adopt a two-stage vulnerability introduction discovery technique and comprehensive evaluation methodology. We prioritize realism and complexity, offering a high-fidelity evaluation environment that previous benchmarks often lack. Moreover, compared with the similar work SecRepoBench [5], we actually include a larger number of projects (41 vs. 27), even though SecRepoBench only considers code completion tasks.
>
> References:
>
> [1] https://www.zdnet.com/article/microsoft-70-percent-of-all-security-bugs-are-memory-safety-issues/
>
> [2] The Chromium Projects. "Memory Safety." Chromium Security, 2025, https://www.chromium.org/Home/chromium-security/memory-safety/.
>
> [3] The White House. "Press Release: Future Software Should Be Memory Safe." Office of the National Cyber Director, 26 Feb. 2024, https://bidenwhitehouse.archives.gov/oncd/briefing-room/2024/02/26/press-release-technical-report/.
>
> [4] Szekeres, Laszlo, et al. "SoK: Eternal war in memory." 2013 IEEE Symposium on Security and Privacy. IEEE, 2013.
>
> [5] Dilgren, Connor, et al. "SecRepoBench: Benchmarking LLMs for secure code generation in real-world repositories." arXiv preprint arXiv:2504.21205 (2025).

---

> ### Author Response · Authors · 2025-11-22
> **Official Response of Submission460 by Authors (2/N)**
>
> **Q2**
> > To point iii) above; there is a certain potential concern of data contamination, which should be discussed/treated with care.
>
> **A2**
>
> We acknowledge the valid concern about potential data contamination. While we cannot completely rule out this possibility due to the opacity of proprietary LLMs' training data, we have taken several measures to mitigate its impact:
>
> 1. Historical commits vs. pre-training data: We adopt historical commits of these repositories on GitHub, but generally pre-training data of large language models only contains the latest version of repositories, instead of the full commit history. For example, the technical report of The Stack V2 [1], a widely adopted dataset for pre-training code LLMs, states that they only extract the latest revision of repositories. Since VICs are typically not the latest version of a project (as there are always fixing commits after VICs in our dataset), there is a high probability that our tasks are not included in the training data. We collected the knowledge cutoff dates [2] and VIC timestamps into the following table, which shows that the latest VIC predates all knowledge cutoff dates:
>
> | GPT-4.1 | Claude Sonnet 3.7 | DeepSeek V3.1 | Latest Time of VIC |
> |---------|-------------------|---------------|-------------------|
> | 06/2024 | 11/2024 | *No earlier than 12/2024 | 03/2024 (Instance 67231) |
>
> *DeepSeek does not release the knowledge cutoff date of DeepSeek V3.1. We infer that its knowledge cutoff is equal to or later than that of DeepSeek V3 (i.e., 12/2024).
>
> 2. Synthesized task descriptions: The task descriptions are synthesized from multiple data sources rather than directly using original information. Therefore, these requirements are unfamiliar to the models and agents. Additionally, considering the high task complexity (i.e., cross-file code editing in large-scale codebases), we argue that simple memorization would hardly help these models and agents solve these problems, which require strong code reasoning abilities. The unsatisfactory experimental results (with a maximum resolved rate of 15.2%) also support this argument.
>
> 3. Evidence from similar benchmarks: Two recent benchmarks on LLM agents for cybersecurity, CyberGym [3] and SEC-bench [4], both demonstrated that there is no statistical difference between results on tasks before and after the knowledge cutoff dates. Given the domain and task similarity, we believe the impact of potential data contamination on our evaluation results is limited.
>
> Despite these mitigation measures, we acknowledge that it is difficult to completely avoid data contamination due to factors such as the lack of transparency in proprietary LLMs and the black-box nature of LLM reasoning. We will add these discussions to our revised version to strengthen our paper.
>
> References:
>
> [1] Lozhkov, Anton, et al. "Starcoder 2 and the stack v2: The next generation." arXiv preprint arXiv:2402.19173 (2024).
>
> [2] Wang, Haooo. "llm-knowledge-cutoff-dates." GitHub, 2025, github.com/HaoooWang/llm-knowledge-cutoff-dates.
>
> [3] Wang, Zhun, et al. "CyberGym: Evaluating AI Agents' Cybersecurity Capabilities with Real-World Vulnerabilities at Scale." arXiv preprint arXiv:2506.02548 (2025).
>
> [4] Lee, Hwiwon, et al. "SEC-bench: Automated Benchmarking of LLM Agents on Real-World Software Security Tasks." NeurIPS 2025.

---

> ### Author Response · Authors · 2025-11-22
> **Official Response of Submission460 by Authors (3/N)**
>
> **Q3**
> > Why is BaxBench excluded from the comparison in Table 1? How does it compare?
> > From the statistics in Table 2, it is clear that the benchmark focuses on small amount of generated code in large, complex contexts. While this is akin to other agentic coding benchmarks (e.g., SWE-Bench) this should be explicitly mentioned in the comparisons with prior secure coding benchmarks (e.g., BaxBench requires the models to generate up to hundreds of lines of code across several files from scratch)
>
> **A3**
>
> We are going to revise Table 1 to include BaxBench. It uses repository-level code generation tasks with functional and security oracles. However, it does not use real historical vulnerability introduction coding scenarios, and ignores the detection of new potential security issues.
>
> BaxBench is an excellent work that motivates us, however, there are the following differences between BaxBench and SecureAgentBench:
>
> 1. BaxBench synthesizes artificial scenarios for secure coding, while SecureAgentBench utilizes real vulnerabilities in different projects, where we can see if agents will introduce vulnerabilities that human developers introduced in the past.
>
> 2. BaxBench focuses on software development, i.e., implementing the requirements from scratch, whereas SecureAgentBench pays attention to the evolution stage, where developers need to maintain existing code bases. Therefore, we believe they are orthogonal to each other due to the difference of coding scenarios.
>
> 3. BaxBench does not consider possibly new vulnerabilities that could be introduced besides security exploits that can be detected, while SecureAgentBench adds SAST to mitigate this issue.
>
> In our revised version, we will incorporate the discussion into the Related Work section to position our work more clearly in previous research. We will also explicitly mention the scenario and task difference to make a fairer comparison with related works.

---

> ### Author Response · Authors · 2025-11-22
> **Official Response of Submission460 by Authors (4/N)**
>
> **Q4**
> > I find the evaluation section, given that this is a benchmark paper, rather limited, mostly due to the lack of actual state-of-the-art systems being evaluated. Giving weight to my concerns are some counterintuitive findings, e.g., figure 5, showing that cheaper models are better. This suggests to me that the benchmark might be noisy for these lower-performant models, and results on stronger agents+models would be more insightful (e.g., Codex + GPT5, Claude Code + Claude Sonnet 4.5).
>
> **A4**
>
> We collect the results of popular coding benchmarks (i.e., SWE-bench, Aider Plotglot) for our subject models. These two benchmarks are widely adopted by leading AI companies, such as OpenAI and Anthropic, to demonstrate the software engineering abilities of LLMs. Here we would like to make a simple comparison between the performances of different benchmarks. Overall, we find that the performance of models on our SecureAgentBench is positively correlated on the whole, e.g., DeepSeek-V3.1 shows consistent superiority on all three benchmarks over Claude 3.7 Sonnet and GPT-4.1. We also provide the price of the three models, and we find that although DeepSeek-V3.1 shows great performance, its cost per 1M tokens is the lowest. These discrepency may rise from different business strategies of different companies. Therefore, we think that the results of Figure 5 are somewhat reasonable considering the aforementioned analysis.
>
> | Model | Release Time | SWE-bench Verified | Aider Plotglot | SecureAgent Bench (Best) | Cost Per 1M Tokens (Input / Output) $ [4] |
> |-------|--------------|-------------------|----------------|--------------------------|------------------------------------------|
> | DeepSeek-V3.1 [1] | 08/2025 | 66.0% (internal code agent) | 68.4% | 15.2% (SWE) | 0.28 / 0.42 |
> | Claude 3.7 Sonnet [2] | 02/2025 | 52.8% (mini-SWE-agent) / 62.3% (agentless) / 70.3% (custom scaffold) | 64.9% | 9.5% (OH) | 3 / 15 |
> | GPT-4.1 [3] | 04/2025 | 54.6% (mini-SWE-agent) / 55.0% (internal) | 52.4% | 9.5% (OH) | 3 / 12 |
>
> Regarding new baselines, we are currently running the following experiments: 1) adding Claude Sonnet 4.5 and GPT-5 into our main experiments (with scaffolds including SWE-agent, OpenHands, and Aider); 2) testing the performance of recent novel techniques including Claude Code + Claude Sonnet 4.5 and Codex + GPT-5. Our preliminary results show that SWE-agent w/ Claude Sonnet 4.5 can achieve a around 39% resolve rate on SecureAgentBench, a surprising improvement compared with existing baselines. We will update our results once we finish these experiments.
>
> References:
>
> [1] Reference for DeepSeek-V3.1: https://huggingface.co/deepseek-ai/DeepSeek-V3.1, Models & Pricing | DeepSeek API Docs
>
> [2] Reference for Claude Sonnet 3.7: Claude 3.7 Sonnet and Claude Code \ Anthropic, SWE-bench Leaderboards,  Aider LLM Leaderboards | aider, Pricing - Claude Docs
>
> [3] Reference for GPT-4.1: Introducing GPT-4.1 in the API | OpenAI, SWE-bench Leaderboards, Aider LLM Leaderboards | aider, Pricing | OpenAI
>
> [4] For input, we use the price when the cache is not cached because the cached input price calculation depends on time for Claude 3.7 Sonnet.

---

> ### Author Response · Authors · 2025-11-22
> **Official Response of Submission460 by Authors (5/N)**
>
> **Q5**
> > Additionally, I miss two critical further examinations of the SAST results: i) the ratio of correct but suspicious programs is rather high, but is not examined further; ii) it seems to me that the SAST result post-patch are not baselined by SAST results pre-patch and SAST results post-goldpatch. Both of these experiments would give us a clearer view of the actual ratio of new vulnerabilities introduced by the agents. Finally, I am missing a more detailed discussion on the rather surprising stark differences in the overall performance and failure patterns of the agent scaffoldings. It would be nice to link the observed performances to design choices in the agents.
>
> **A5**
>
> 1. High ratio of suspicious programs. We agree this deserves deeper investigation, and in our current version of manuscript, we did a brief discussion about them (“Newly Introduced Security Risks”). In the revision, we will add a detailed part analyzing the results of SAST to improve our evaluation analysis section.
>
> 2. SAST baseline comparison. We appreciate this very insightful feedback and want to clarify our design choice. We opted to compare post-agentpatch against pre-patch (rather than against post-goldpatch) for the following reason: our goal is to measure the absolute security risks introduced by agents, not the relative difference from gold patches. Even if a gold patch contains certain vulnerabilities, any new security issue introduced by the agent still represents a real risk to users. Therefore, it seems more reasonable to measure the absolute risks and report this kind of results.
>
> 3. Failure mode. We will add a new subsection discussing the failure modes of different agents and models, to make a more detailed analysis about how agent fails in secure coding.
>
> **Q6**
> > Minor: Presentation. I find some of the tables and figures hard/strenuous to read. Table5: uncomfortable to match the middle column to the left column---I suggest a clearer horizontal divide between the CWEs. Tables 2 and 3 are next to eachother but misaligned, creating a poor visual effect, same for Table 5 and Figure 6. Figure 4 is overloaded---I suggest focusing in the main experiment on less but higher-level metrics (e.g., C&S, CS, and a larger category of incorrect), and then have a subsequent analysis of the errors. It makes for a clearer presentation. Table 4 is dense and hard to follow the column alignments. Figure 5 violates the page formatting, I believe this is an honest error, but I advise the authors to fix it at the next possible chance. Finally, the paper has minor typos and grammatical errors that could be quickly fixed by running a spell-checker or other writing aids.
>
> **A6**
>
> We thank very much for the detailed suggestions about our presentation. We will fix and improve them in the revised manuscript.
>
> **Q7**
> > It seems that SecRepoBench already contains most of the main assets needed to construct SecureAgentBench. Why was a new data collection pipeline necessary? Could you not have just extended SecRepoBench (and potentially other prior works)?
>
> **A7**
>
> SecRepoBench is an excellent work for evaluating the secure coding capability of LLMs using function completion tasks. However, although we both utilize OSS-Fuzz as the data source, our data collection pipeline differs significantly from theirs in the following perspectives:
>
> 1. Functional oracle: In their work, the level of functional test is method-level, i.e., they only require the unit test of one specific function. However, we utilize the full test suite (beyond only unit tests) of the whole repository, which is non-trivial as it requires a lot of manual work to 1) build and run the test suite of the project, and 2) write and verify the scripts to parse the results of different test frameworks of different projects. We devoted substantial manual efforts to accomplish these oracles.
>
> 2. Vulnerability introduction context: SecRepoBench masks the region of vulnerability fixing and takes the surrounding code as the context of code completion, which is misaligned with the scenario where human developers introduced vulnerabilities in the past. To the best of our knowledge, no previous works adopt a vulnerability introduction context of real vulnerabilities. This distinction is fundamental: we evaluate whether agents can avoid introducing vulnerabilities in realistic development scenarios, not whether they can patch known vulnerabilities.
>
> 3. Availability: At the time of our data collection, SecRepoBench's code and data were not publicly available. While they plan to release it, we needed to establish our own validated pipeline to ensure reproducibility and enable community extension of our benchmark.
>
> Beyond these methodological differences, SecureAgentBench focuses on “vibe coding” scenarios, which require a different evaluation framework than function-level completion tasks. We view our work as complementary to SecRepoBench, targeting different aspects of LLM security evaluation.

---

> ### Author Response · Authors · 2025-11-22
> **Official Response of Submission460 by Authors (6/N)**
>
> **Q8**
> > While the at many stages of the benchmark construction pipeline manual intervention is needed, the task descriptions are LLM-generated. Why so? It seems that this would have been a smaller manual effort then for instance verifying and setting up the functional testing suites.
>
> **A8**
>
> For the task descriptions, we first use LLMs to generate the requirements based on diverse contexts (e.g., gold patches, commit messages, related issues and discussions), and then humans conduct manual inspection to ensure the quality of the benchmark. We agree that it is straightforward to manually write the task descriptions, but we chose the current solution for these reasons:
>
> 1. Unlike building tests or writing parsers by annotators, which have objective standards to verify correctness, problem description writing is somewhat subjective and easily affected by human bias. LLMs prompted with the same instruction can mitigate this issue and help maintain description consistency across different tasks in terms of style, format, and abstraction level (while remaining security-neutral).
>
> 2. We leverage multiple information sources including code patches, commit messages, issue descriptions, discussions, etc. LLMs can effectively integrate diverse data sources while following instructions such as maintaining security neutrality. This significantly reduces the manual effort required to generate descriptions, while ensuring quality through further manual inspection.
>
> 3. Previous works [1-3] such as BigCodeBench and ToolBench have proven the effectiveness of human-AI collaboration in building benchmarks, and we follow this construction paradigm to achieve a balance between efficiency and quality.
>
> In summary, our approach leverages LLM generation combined with human inspection to balance quality and efficiency in problem description construction, which has been proven to be an effective paradigm in previous works. We acknowledge the potential bias that could be introduced by the process of getting descriptions, and we plan to add discussion about this limitation in our revised paper.
>
> References:
>
> [1] Zhuo, Terry Yue, et al. "Bigcodebench: Benchmarking code generation with diverse function calls and complex instructions." ICLR. 2025.
>
> [2] Qin, Yujia, et al. "ToolLLM: Facilitating Large Language Models to Master 16000+ Real-world APIs." ICLR. 2024.
>
> [3] Chen, Haorui, Chengze Li, and Jia Li. "FeatBench: Evaluating Coding Agents on Feature Implementation for Vibe Coding." arXiv preprint arXiv:2509.22237 (2025).
>
> **Q9**
> > The data collection pipeline is rather lossy. Starting from almost 5k potential cases, at the end only 105 remain. What do you think, where could you improve your pipeline to decrease this loss? Could you automate more steps to have potentially a constantly evolving benchmark, reducing, e.g., contamination risks?
>
> **A9**
>
> We acknowledge the significant loss of data instances, and the primary cause is the accurate location of vulnerability-inducing commit (VIC) (i.e., 4,993 -> 254). We utilize two stages to strictly assure the quality of vulnerability introduction context, as it is essential in our research to mimic the scenarios where human developers introduced real vulnerabilities in the past.
>
> Due to the above reason, if we can find accurate VICs for more instances, we can lose fewer vulnerability cases. However, unfortunately, the current research on VIC discovery reveals that the accurate selection of VICs is very difficult, and although some approaches (i.e., SZZ alone) can be applied to more instances, their accuracy is far from satisfactory, as described in the paper, which decreases the reliability and quality of our benchmark.
>
> We agree that some steps (e.g., vulnerability collection) could be automated, and we think it is a great direction for future work to explore. We will add this discussion to our paper to make it more comprehensive.
>
>
> **Q10**
> > For the results in Figure 7, this seems rather counterintuitive. What happens if you increase the budget constraints? What happens if you try better models/scaffolds? Are the 16 C&S cases the same or different with and without the prompt?
>
> **A10**
>
> Thank you for the constructive feedback. We are planning to experiment with a more comprehensive analysis about the security reminder (considering the aforementioned factors), and we will update once it is finished.

---

> > ### Comment · Reviewer_hdwE · 2025-11-24
> >
> > Thank you for the detailed rebuttal!
> >
> > The planned changes and preliminary results look promising, however, I will wait until they are done and implemented in the paper before I reevaluate my initial judgement.
> >
> > Moreover, I have a follow-up comment to Q5-A5:
> >
> > > I understand that the total newly introduced vulnerabilities are the most significant, however, I still believe that adding the suggested baselines would make for a more informative picture as: (i) it would enable the benchmarking of the agents' performance against the human gold patches, giving a clear comparison on where the agents stand; and (ii) would enable to at least a bit denoise the SAST results which likely contain also some false positives/negatives.

---

> ### Author Response · Authors · 2025-11-25
>
> Thank you very much for your timely response and for acknowledging our revision plan! We will update the results as soon as they are complete, and we sincerely appreciate your understanding.
>
> > I understand that the total newly introduced vulnerabilities are the most significant, however, I still believe that adding the suggested baselines would make for a more informative picture as: (i) it would enable the benchmarking of the agents’ performance against the human gold patches, giving a clear comparison on where the agents stand; and (ii) would enable to at least a bit denoise the SAST results which likely contain also some false positives/negatives.
>
> We fully agree with your insights, and we appreciate the clarity of your reasoning. This is a promising direction for helping to clean and filter existing SAST results, and makes it easier to understand the gap between humans and agents. In response, we have decided to introduce this new “suspicious” setting to explicitly examine the delta between human patches and agent-generated patches with respect to SAST results. Additional analyses and discussions will be incorporated into the revised paper.
>
> Thank you again for your valuable feedback and your supportive understanding.

---

> > ### Comment · Reviewer_hdwE · 2025-11-25
> >
> > Looking forward!

---

### Official Review · Reviewer_iUa5 · 2025-10-27

**Soundness:** 2
**Presentation:** 3
**Contribution:** 2
**Rating:** 4
**Confidence:** 4

**Summary:**

This paper introduces SecureAgentBench, a benchmark that evaluates whether an Agent can implement functionalities with secure code. They prepare 105 coding tasks from the data source ARVO and OSS-Fuzz to make sure each task includes realistic multi-file edit requirements and contains potential vulnerabilities from human implementation. For the evaluation metric, they consider both to evaluate the functionality and the security (poc and static analysis). They test open-source agents (SWE-agent, OpenHands, and Aider) with SOTA models on this benchmark, including GPT, Claude, and Deepseek.

**Strengths:**

1. Promising direction. Establishing the benchmark to highlight the security risks associated with Code GenAI is a direction worth studying.
2. Consider real-world security problems in OSS repo and environment deployments for agentic tasks.
3. Compared with existing baselines from multiple perspectives, the results show the effectiveness of the proposed method.
4. Consider multiple metrics for evaluating code patches generated from the agent. For example, using SAST to illustrate the consideration of the robustness of security.

**Weaknesses:**

1. Currently, the benchmark only contains C/C++ language data, which might be considered insufficient for comprehensive evaluation. And the data collection method should be considered, whether it can be automatic or semi-automatic for continuous collection.
2. There are some concerns about the data collection and evaluation that need to be clarified. Generally, there would be more than 1 commit from PVIC/VIC to VFC. For the task description, what would be involved: only the VIC commit context, or from PVIC to VFC? What version of unit test are you using for the functionality test? Does it consist with the task description? For example, if you are asking an agent to implement the functionality of VIC but using the unit test from VFC, some cases might never be considered correct. How can we guarantee the quality of the task description if it is generated by an LLM?

**Questions:**

1. As we already use SAST for evaluation, it would be great to show the important rules you are running in the tool. Otherwise reader still does not know what this test covers.

2. Have you considered data contamination for the data collection? How can we guarantee the LLMs are not that familiar with the repo and code?

3. I am curious about why Aider has such amount of "empty patch" in the results. It would be good to have a case study on why these happen to exclude the possibilities of misoperation.

4. Unit test: what we mentioned above.

5. It might be good to also provide the time and cost of the experiments for a better understanding of the difficulty of the task.

6. It might be good to cite some rule-based patching agents, even though they are not evaluated in the experiments.

[1] Agentless: Demystifying llm-based software engineering agents

[2] PatchPilot: A Cost-Efficient Software Engineering Agent with Early Attempts on Formal Verification

---

> ### Author Response · Authors · 2025-11-22
> **Official Response of Submission460 by Authors (1/N)**
>
> We thank you so much for your constructive and valuable feedback. The following is our reponse:
>
> **Q1**
> > Currently, the benchmark only contains C/C++ language data, which might be considered insufficient for comprehensive evaluation. And the data collection method should be considered, whether it can be automatic or semi-automatic for continuous collection.
>
> **A1**
>
> 1. Language: We acknowledge the limitation regarding the language scope. In our revised version, we will explicitly present that our benchmark is focused on C/C++. Besides, we want to highlight that C/C++ remains the cornerstone of critical infrastructure, making its vulnerabilities highly significant and the subject of intense scrutiny from both industry and government bodies [1-4]. Furthermore, compared to simpler benchmarks, our work prioritizes realistic scenarios and a comprehensive evaluation, addressing the vulnerability introduction and repository-level editing tasks that previous studies often overlook.
>
> 2. Automation. In our data collection pipeline, we adopt human efforts to acquire functional tests and ensure the quality of natural language requirements, in order to make a high-quality evaluation benchmark. However, we fully agree that there is great potential for (semi-)automatic continuous collection with techniques such as test generation techniques. We will add these discussions about limitations and future work into our revised version of the paper to make it sound robust.
>
> References:
>
> [1] https://www.zdnet.com/article/microsoft-70-percent-of-all-security-bugs-are-memory-safety-issues/
>
> [2] The Chromium Projects. “Memory Safety.” Chromium Security, 2025, https://www.chromium.org/Home/chromium-security/memory-safety/.
>
> [3] The White House. “Press Release: Future Software Should Be Memory Safe.” Office of the National Cyber Director, 26 Feb. 2024, https://bidenwhitehouse.archives.gov/oncd/briefing-room/2024/02/26/press-release-technical-report/.
>
> [4] Szekeres, Laszlo, et al. "Sok: Eternal war in memory." 2013 IEEE Symposium on Security and Privacy. IEEE, 2013.
>
> **Q2**
> > There are some concerns about the data collection and evaluation that need to be clarified. Generally, there would be more than 1 commit from PVIC/VIC to VFC. For the task description, what would be involved: only the VIC commit context, or from PVIC to VFC? What version of unit test are you using for the functionality test? Does it consist with the task description? For example, if you are asking an agent to implement the functionality of VIC but using the unit test from VFC, some cases might never be considered correct. How can we guarantee the quality of the task description if it is generated by an LLM?
>
> **A2**
>
> 1. For task description and evaluation oracle (of both functionality and security), we consider one commit (i.e., VIC) instead of several commits, such as PVIC to VFC. By doing this, we can create the coding scenario where human developers introduced vulnerabilities (that is, the vulnerability introduction scenario), and the task description is consistent with the commit and evaluation oracle.
>
> 2. We employ a multi-layered approach to guarantee task description quality: 1) Structured prompting: We use carefully designed prompt templates to ensure descriptions are security-neutral and clear; 2) Diverse data sources: We incorporate multiple information sources (e.g., commit messages, code diffs, issue trackers) to make descriptions comprehensive and informative; 3) Human verification: All generated descriptions undergo final quality verification by human experts to ensure the quality of these requirements. This pipeline helps us maintain high-quality, reliable task descriptions despite using LLM generation.
>
> **Q3**
> > As we already use SAST for evaluation, it would be great to show the important rules you are running in the tool. Otherwise reader still does not know what this test covers.
> > It might be good to cite some rule-based patching agents, even though they are not evaluated in the experiments.
>
> **A3**
>
> Thank you very much for the advice. In our revised version of the paper, we will 1) list detailed rules that SAST leverages to scan the codebase; 2) add these related works about rule-based code agents to make our paper more complete and comprehensive.

---

> ### Author Response · Authors · 2025-11-22
> **Official Response of Submission460 by Authors (2/N)**
>
> **Q4**
> > Have you considered data contamination for the data collection? How can we guarantee the LLMs are not that familiar with the repo and code?
>
> **A4**
>
> We acknowledge the valid concern about potential data contamination. While we cannot completely rule out this possibility due to the opacity of proprietary LLMs' training data, we have taken several measures to mitigate its impact:
>
> 1. Historical commits vs. pre-training data: We adopt historical commits of these repositories on GitHub, but generally pre-training data of large language models only contains the latest version of repositories, instead of the full commit history. For example, the technical report of The Stack V2 [1], a widely adopted dataset for pre-training code LLMs, states that they only extract the latest revision of repositories. Since VICs are typically not the latest version of a project (as there are always fixing commits after VICs in our dataset), there is a high probability that our tasks are not included in the training data. We collected the knowledge cutoff dates [2] and VIC timestamps into the following table, which shows that the latest VIC predates all knowledge cutoff dates:
>
> | GPT-4.1 | Claude Sonnet 3.7 | DeepSeek V3.1 | Latest Time of VIC |
> |---------|-------------------|---------------|-------------------|
> | 06/2024 | 11/2024 | *No earlier than 12/2024 | 03/2024 (Instance 67231) |
>
> *DeepSeek does not release the knowledge cutoff date of DeepSeek V3.1. We infer that its knowledge cutoff is equal to or later than that of DeepSeek V3 (i.e., 12/2024).
>
> 2. Synthesized task descriptions: The task descriptions are synthesized from multiple data sources rather than directly using original information. Therefore, these requirements are unfamiliar to the models and agents. Additionally, considering the high task complexity (i.e., cross-file code editing in large-scale codebases), we argue that simple memorization would hardly help these models and agents solve these problems, which require strong code reasoning abilities. The unsatisfactory experimental results (with a maximum resolved rate of 15.2%) also support this argument.
>
> 3. Evidence from similar benchmarks: Two recent benchmarks on LLM agents for cybersecurity, CyberGym [3] and SEC-bench [4], both demonstrated that there is no statistical difference between results on tasks before and after the knowledge cutoff dates. Given the domain and task similarity, we believe the impact of potential data contamination on our evaluation results is limited.
>
> Despite these mitigation measures, we acknowledge that it is difficult to completely avoid data contamination due to factors such as the lack of transparency in proprietary LLMs and the black-box nature of LLM reasoning. We will add these discussions to our revised version to strengthen our paper.
>
> References:
>
> [1] Lozhkov, Anton, et al. "Starcoder 2 and the stack v2: The next generation." arXiv preprint arXiv:2402.19173 (2024).
>
> [2] Wang, Haooo. "llm-knowledge-cutoff-dates." GitHub, 2025, github.com/HaoooWang/llm-knowledge-cutoff-dates.
>
> [3] Wang, Zhun, et al. "CyberGym: Evaluating AI Agents' Cybersecurity Capabilities with Real-World Vulnerabilities at Scale." arXiv preprint arXiv:2506.02548 (2025).
>
> [4] Lee, Hwiwon, et al. "SEC-bench: Automated Benchmarking of LLM Agents on Real-World Software Security Tasks." NeurIPS 2025.
>
> **Q5**
> > I am curious about why Aider has such amount of "empty patch" in the results. It would be good to have a case study on why these happen to exclude the possibilities of misoperation.
>
> **A5**
>
> We acknowledge the substantial proportion of "No Output" cases, particularly for the Aider agent. This phenomenon has also been observed in recent cybersecurity agent research, such as SEC-Bench [1] (e.g., 40 “No Output” out of 71 failures), which attributes it to context window limitations and agent framework constraints. To provide deeper insights, we will include a failure mode analysis in the revised manuscript, examining the potential causes across different agent-model combinations.
>
> References:
>
> [1] Lee, Hwiwon, et al. "SEC-bench: Automated Benchmarking of LLM Agents on Real-World Software Security Tasks." NeurIPS. 2025.
>
> **Q6**
> > It might be good to also provide the time and cost of the experiments for a better understanding of the difficulty of the task.
>
> **A6**
>
> In the results analysis of our paper, we include a discussion section on the cost of the experiments. In our revised version, we will add more detailed analyses about time and cost, including new experiements with agents and models such as Claude Code and GPT-5.

---

### Official Review · Reviewer_hP6r · 2025-11-01

**Soundness:** 3
**Presentation:** 3
**Contribution:** 2
**Rating:** 4
**Confidence:** 4

**Summary:**

This paper introduces a new benchmark, SecureAgentBench, a benchmark of 105 coding tasks designed to evaluate code agents’ capabilities in secure code generation. The coding tasks are sourced from real-world software repositories, undergoing rigorous data cleaning and filtering process. The code patches generated by the code agents are evaluated from various aspects including the correctness of the functionality, whether it contains already known vulnerabilities, and whether it introduces new vulnerabilities. The evaluation results demonstrate that existing code agents struggle to produce correct and secure code.

**Strengths:**

- The benchmark requires code agents to operate directly on entire software repositories rather than on simplified, function-level completion tasks. This design better reflects how agents perform in realistic software engineering workflows.
- The data collection, cleaning, and filtering pipeline is rigorous, ensuring the benchmark’s reliability and high-quality task instances.
- The paper presents a comprehensive evaluation across multiple agent frameworks, backbone models, and experimental settings (including the effect of explicit security reminders), offering a thorough analysis of current capabilities and limitations.

**Weaknesses:**

- Many real-world, repository-level feature implementations require multiple commits, whereas this benchmark considers only a single commit per task. The gold patches average around 40 lines of code across two files, which limits the overall task complexity.
- The process for collecting and validating test cases is not clearly described. If the tests are sourced from PVIC, they may not adequately verify the newly implemented feature. Conversely, if they are derived from later commits, it remains unclear how these test cases are identified and extracted.
- The benchmark size is relatively small, with only 105 tasks, which may restrict the statistical robustness of the evaluation.
- The benchmark exhibits potential bias and limited coverage for code generation tasks, sourcing from commits introducing C/C++ memory-safety vulnerabilities historically found by fuzzing.
- The analysis of SAST results lacks sufficient discussion of false positives, which could affect the reliability of the reported “suspicious” vulnerability cases.

**Questions:**

Check weaknesses.

---

> ### Author Response · Authors · 2025-11-22
> **Official Response of Submission460 by Authors (1/N)**
>
> Thank you so much for the detailed feedback on our manuscript. The following are our reponses:
>
> **Q1**
> > Many real-world, repository-level feature implementations require multiple commits, whereas this benchmark considers only a single commit per task. The gold patches average around 40 lines of code across two files, which limits the overall task complexity.
>
> **A1**
> 1. Single commit. We emphasize the importance of vulnerability introduction context because it allows us to reproduce the programming scenarios where human developers introduced real vulnerabilities in the past. Therefore, we need to backtrace to the exact version that induced the security issue. We consider a single commit to ensure the quality of VIC identification through our two-stage approach, which is critical to achieving our target scenario.
>
> 2. Comparison with similar works. We collected statistics from similar software engineering and cybersecurity benchmarks in the following table, such as SWE-bench and CyberGym [1-3]. These benchmarks have been proven to be difficult and demanding for current code agents and LLMs, and our benchmark presents comparable statistics.
>
> 3. Complexity of SecureAgentBench. Although the statistics of gold patches in these tasks may not appear large, the tasks still require deep code understanding and reasoning abilities to comprehend large-scale repositories (averaging 554K LOC and 2.8K files) and requirements (averaging 200 words), perform correct localization and implementation, and pass comprehensive evaluations. The unsatisfactory performance of different agents and LLMs (i.e., up to 15.2% resolve rate) also reflects the high complexity.
>
> | Task | Avg. # of lines of code | Avg. # of files |
> |------|------------------------|-----------------|
> | CyberGym [1]  | 7 (Median) | 1 (Median) |
> | SEC-Bench [2] | 17.3 | 1.3 |
> | SWE-Bench [3] | 32.8 | 1.7 |
> | **SecureAgentBench (Ours)** | **42.5** | **1.9** |
>
> References:
>
> [1] Wang, Zhun, et al. "CyberGym: Evaluating AI Agents' Cybersecurity Capabilities with Real-World Vulnerabilities at Scale." arXiv preprint arXiv:2506.02548 (2025).
>
> [2] Lee, Hwiwon, et al. "SEC-bench: Automated Benchmarking of LLM Agents on Real-World Software Security Tasks." NeurIPS. 2025.
>
> [3] Jimenez, Carlos E., et al. "SWE-bench: Can Language Models Resolve Real-world Github Issues?." ICLR. 2024.
>
>
>
> **Q2**
> > The process for collecting and validating test cases is not clearly described. If the tests are sourced from PVIC, they may not adequately verify the newly implemented feature. Conversely, if they are derived from later commits, it remains unclear how these test cases are identified and extracted.
>
> **A2**
>
> 1. Source of test case. Our functional tests are extracted from the repository's official test suite at the VIC version. We focus on evaluating whether agent-generated patches achieve equivalent functionality to the gold patch under the VIC context by differential testing. To accomplish this, we use the complete test suite from VIC, avoiding the complexity and potential bias of selectively extracting specific test cases, and statistics in Table 2 demonstrate the quality of our test oracles (i.e., about 434 test cases on average). To obtain these tests and their corresponding parsers, we manually write scripts for each task to compile, build, run, and parse test. It requires substantial efforts but ensure reliable and accurate functionality evaluation.
>
> 2. Later commits. We deliberately do not consider test cases from commits after VIC, as agents are tasked with implementing the functionality at the VIC context, not subsequent commits. Using later tests would misalign the evaluation context and potentially introduce requirements beyond the task scope.
>
> 3. Presentation improvement. In our revised version of manuscript, we plan to improve the presentation and statement of rationales and implementations about functionality test oracle in both main text and appendix.

---

> ### Author Response · Authors · 2025-11-22
> **Official Response of Submission460 by Authors (2/N)**
>
> **Q3**
> > The benchmark size is relatively small, with only 105 tasks, which may restrict the statistical robustness of the evaluation.
>
> **A3**
>
> We acknowledge this limitation regarding benchmark size. However, we would like to clarify from the following aspects:
>
> 1. Task granularity. Our benchmark adopts SWE-bench-style repository-level code editing tasks, which are more challenging than related works that use code completion tasks. Despite this higher complexity, we still achieve a comparable scale to widely adopted works such as CWEval [1], which consists of 119 coding tasks.
>
> 2. Quality and comprehensiveness. We emphasize quality than scale to ensure statistical validity, so we employ a hybrid approach to discover accurate VICs, implement a strict data filtering pipeline, and utilize both dynamic and static oracles. In our evaluation, we provide fine-grained analysis including different failure modes, cost, and vulnerability types. We also observe statistically meaningful trends, as evidenced by consistent patterns across different agents, models, and vulnerability types.
>
> 3. Future work. We plan to discuss the extensibility of our approach and plans for expansion in the limitation section.
>
> References:
>
> [1] Peng, Jinjun, et al. "Cweval: Outcome-driven evaluation on functionality and security of llm code generation." 2025 IEEE/ACM International Workshop on Large Language Models for Code (LLM4Code). IEEE, 2025.
>
>
> **Q4**
> > The benchmark exhibits potential bias and limited coverage for code generation tasks, sourcing from commits introducing C/C++ memory-safety vulnerabilities historically found by fuzzing.
>
> **A4**
>
> We acknowledge the limitations regarding programming language diversity and vulnerability type coverage in our benchmark. We plan to explicitly discuss and clearly state this scope in the revised version to make our focus more transparent.
>
> Our work specifically targets memory-safety issues in C/C++, as C/C++ remains the most critical programming language in modern digital infrastructure. Memory-safety vulnerabilities have been reported as the most severe security concerns by major organizations such as Microsoft [1, 2] and the Chrome team [3]. We are committed to providing a realistic and challenging benchmark for evaluating code agents in these critical scenarios, which distinguishes our work from previous benchmarks.
>
> References:
>
> [1] Szekeres, Laszlo, et al. "SoK: Eternal War in Memory." 2013 IEEE Symposium on Security and Privacy. IEEE, 2013.
>
> [2] "Microsoft: 70 Percent of All Security Bugs Are Memory Safety Issues." ZDNet, https://www.zdnet.com/article/microsoft-70-percent-of-all-security-bugs-are-memory-safety-issues/.
>
> [3] The Chromium Projects. "Memory Safety." Chromium Security, 2025, https://www.chromium.org/Home/chromium-security/memory-safety/.
>
>
>
>
>
> **Q5**
> > The analysis of SAST results lacks sufficient discussion of false positives, which could affect the reliability of the reported “suspicious” vulnerability cases.
>
> **A5**
> We acknowledge that SAST tools can produce false positives, which is precisely why we label such cases as "suspicious" rather than "vulnerable." However, we believe this approach remains acceptable for the following reasons:
>
> 1. Oracles. Our security evaluation employs a multi-faceted approach that combines both SAST and PoC exploits. These two methods are complementary: PoC provides high-confidence validation of known vulnerabilities, while SAST helps detect potential new security risks that may not have existing exploits. This dual-oracle design enhances the overall reliability of our security assessment.
>
> 2. SAST selection. To mitigate false positive concerns, we use Semgrep, a widely-adopted SAST tool with over 2k rules (as mentioned in Appendix C). The extensive rule coverage helps improve detection accuracy. Additionally, we distinguish between different confidence levels in our evaluation framework: cases flagged only by PoC are marked as "vulnerable," while those detected solely by SAST are labeled "suspicious," making our classification more rigorous and sound.
>
> 3. Presentation. Furthermore, we will incorporate additional analysis of SAST results in the revised manuscript to provide more transparency about the characteristics and distribution of these findings, helping readers better interpret the "suspicious" category.

---

> > ### Comment · Reviewer_hP6r · 2025-11-23
> >
> > Thank you for the detailed and thoughtful response.
> >
> > **For Q1:**
> > CyberGym and SEC-Bench focus primarily on security patches with PoC generation and patch generation, whereas SecureAgentBench targets secure code generation, so the task nature is slightly different.
> > I understand the motivation for using the VIC and agree that this context can contain more security-sensitive signals relevant for evaluating secure code generation.
> >
> > **For Q2:**
> > since SecureAgentBench requires the agent to generate new code, please clarify how to validate the functionality of these newly implemented features, they don't have corresponding test cases in the previous commits.

---

> > > ### Author Response · Authors · 2025-11-23
> > > **Response for Official Comment by Reviewer hP6r**
> > >
> > > We thanks very much for your prompt reply and feedback! Here is our further clarification:
> > >
> > > **Q1**
> > > > CyberGym and SEC-Bench focus primarily on security patches with PoC generation and patch generation, whereas SecureAgentBench targets secure code generation, so the task nature is slightly different. I understand the motivation for using the VIC and agree that this context can contain more security-sensitive signals relevant for evaluating secure code generation.
> > >
> > > **A1**
> > >
> > > Thank you for acknowledging our motivation and the vulnerability introduction context design. We acknowledge the potential bias in task nature and will add comparative statistics and discussion in our revision to mitigate this issue.
> > >
> > > **Q2**
> > > > since SecureAgentBench requires the agent to generate new code, please clarify how to validate the functionality of these newly implemented features, they don't have corresponding test cases in the previous commits.
> > >
> > > **A2**
> > >
> > > We clarify our functionality test oracle methodology from these points:
> > >
> > > 1. Our approach follows the widely-adopted software engineering practices that require tests to be co-located with code changes in the same commit [1-4]. For example, Google's code review guidance [1] states that "Tests and documentation changes should be in the same CL/commit as the code changes." Similar principles are advocated in popular open-source projects and industry, such as "Test updates needed by a change should be in the same commit as the original change" [3]. As the projects in our benchmark are generally active and well-maintained, we assume they follow these best practices, which means the test suite at each commit should be adequate for validating the corresponding code changes. Therefore, we use the full test suite at VIC as the oracle to evaluate the functionality of agent-generated code.
> > >
> > > 2. We acknowledge that for some tasks introducing entirely new features, VIC tests may not strictly provide comprehensive coverage. However, since we use differential testing to compare post-gold-patch and post-agent-patch behavior, we can ensure a lower bound on patch correctness even in the worst case, i.e., if tests fail, the agent definitively has not produced correct code. In our experiments, the best baseline achieves only ~40% functional correctness, demonstrating both the complexity of our benchmark and the unsatisfactory performance of current agents. To make our paper more sound and complete, we will add discussions about our rationale and limitations in the revised version about this issue.
> > >
> > > References:
> > >
> > > [1] Bartlett, R. A. “Google Guidance on Code Review.” Better Scientific Software, 24 Aug. 2023, bssw.io/items/google-guidance-on-code-review.
> > >
> > > [2] Zulip. “Commit Discipline.” Zulip 11.2 Documentation, version 11.2, readthedocs.io, https://zulip.readthedocs.io/en/11.2/contributing/commit-discipline.html. Accessed 23 Nov. 2025.
> > >
> > > [3] “Code Contribution Best Practices.” moja global Technical Guide, moja global, 2020, Sneha Mishra, docs.moja.global/en/add_instructions_gdal/contributing/code_contribution_best_practices.html.
> > >
> > > [4] “Writing High Quality, Well Scoped, Commits.” CleverThis, 28 Apr. 2025, https://cleverthis.com/news/writing-high-quality-well-scoped-commits/.

---

### Official Review · Reviewer_XBz5 · 2025-11-03

**Soundness:** 2
**Presentation:** 2
**Contribution:** 2
**Rating:** 4
**Confidence:** 3

**Summary:**

The paper introduces SECUREAGENTBENCH, a benchmark designed to evaluate how well LLM-powered code agents can generate secure code. It consists of 105 tasks derived from real-world, open-source vulnerabilities, setting the tasks within the repository context at the exact point the vulnerability was originally introduced. The benchmark evaluates agent-generated patches for both functional correctness (using differential testing) and security (using proof-of-concept exploits and static analysis). Experiments on agents like SWE-agent and OpenHands show they struggle significantly, with the best-performing combination achieving only a 15.2% "correct-and-secure" rate. The results also reveal that agents often introduce new, previously unrecorded vulnerabilities, and that simple security reminders in prompts do not improve performance.

**Strengths:**

- The benchmark addresses the critical and timely issue of secure code generation by agents. SECUREAGENTBENCH requires agents to perform realistic, repository-level edits that often involve multiple files, better simulating real-world software engineering tasks.
- The construction process employs a two-stage pipeline (static SZZ followed by dynamic PoC validation) to identify Vulnerability-Inducing Commits (VIC). This allows tasks to be grounded in the exact historical context where vulnerabilities were originally introduced.
- The work employs a multi-faceted evaluation protocol: 1) Functional Correctness via differential testing against developer reference implementations. 2) Historical Security Verification using specific proof-of-concept (PoC) exploits. 3) New Vulnerability Detection using static analysis (SAST) to flag potential new risks introduced by the agents.

**Weaknesses:**

- Limitations of Security Oracles: The security evaluation relies on PoCs that only detect the single historical vulnerability targeted by each task, combined with a single SAST tool (Semgrep) known for potential false positives and limited by its specific rule set. Consequently, code classified as "secure" by this benchmark may still contain undetected vulnerabilities, preventing the guarantee of security.

- High Generation Failure Rates affecting Insights: The extremely high rate of "No Output" results, such as Aider+GPT failing to produce any output 77.1% of the time, raises questions about the definitive nature of the findings. While the authors attribute this to inherent task difficulty, it might also stem from suboptimal prompt instructions or simply that the evaluated models are currently incapable of handling these benign coding tasks, obscuring their actual security capabilities.
- Unclear Language Coverage and Extensibility: The specific coverage of programming languages is not explicitly defined in the main text, though the prompt templates in Appendix H explicitly specify a "C/C++ code repository". If the dataset is exclusively C/C++, this should be stated more clearly as a scope limitation. Furthermore, it would be beneficial to discuss whether the proposed construction method can be easily extended to other languages beyond C/C++.

**Questions:**

- Ambiguity in Vulnerability Verification: While Section 2.2 explicitly mentions manual quality assurance for test cases and requirements, it is unclear if the specific Vulnerability/CWE classifications underwent human verification to ensure they accurately reflect the root cause of the issues derived from OSS-Fuzz.

---

> ### Author Response · Authors · 2025-11-22
> **Official Response of Submission460 by Authors (1/N)**
>
> We thank the reviewer for this insightful comment and valuable time devoted to this service. Please refer to our clarifications below:
>
> **Q1**
> > Limitations of Security Oracles: The security evaluation relies on PoCs that only detect the single historical vulnerability targeted by each task, combined with a single SAST tool (Semgrep) known for potential false positives and limited by its specific rule set. Consequently, code classified as "secure" by this benchmark may still contain undetected vulnerabilities, preventing the guarantee of security.
>
> **A1**
>
> We organize our response from two perspectives:
> 1. Evaluation oracles. In our paper, we adopt both dynamic (PoC-based) and static (SAST-based) techniques to evaluate the security of agent-generated code. PoC programs verify whether the specific historical vulnerabilities are reintroduced, while SAST detects potential new security risks that were not present in the original codebase with about 26K security rules. As shown in Table 1, our combined approach represents a significant advancement compared with prior works, considering both PoC and SAST. However, we acknowledge that vulnerability detection remains an open and challenging problem· in software security, and no evaluation method can guarantee that code is "completely" secure. In our revised version, we plan to discuss and clarify this limitation about our "secure" classification and possible future directions for other vulnerability detection methods.
> 2. SAST. We are aware of the potential for false positives in SAST tools and the specificity of their rule sets. This is why we explicitly classify SAST-flagged cases as "suspicious" rather than definitively "vulnerable," recognizing that these require further inspection. To strengthen our analysis, we will: (1) add a detailed analysis of SAST results, including false positives in the results, and (2) provide comprehensive documentation of the Semgrep rule set used to ensure transparency about our static analysis coverage.
>
> **Q2**
> > High Generation Failure Rates affecting Insights: The extremely high rate of "No Output" results, such as Aider+GPT failing to produce any output 77.1% of the time, raises questions about the definitive nature of the findings. While the authors attribute this to inherent task difficulty, it might also stem from suboptimal prompt instructions or simply that the evaluated models are currently incapable of handling these benign coding tasks, obscuring their actual security capabilities.
>
> **A2**
>
> 1. No output. We acknowledge the substantial proportion of "No Output" cases, particularly for the Aider agent. This phenomenon has also been observed in recent cybersecurity agent research, such as SEC-Bench [3] (e.g., 40 “No Output” out of 71 failures), which attributes it to context window limitations and agent framework constraints. To provide deeper insights, we will include a failure mode analysis in the revised manuscript, examining the potential causes across different agent-model combinations.
> 2. Experimental setup. Our choices of prompts, LLMs, and agent frameworks are grounded in established practices from prior research:
>     1. Prompt design: We adopted the prompt template from Multi-SWE-Bench [1], which extends the original SWE-Bench [2] to support C/C++ repositories. We made minimal modifications to adapt it from bug-fixing to secure code generation tasks, ensuring consistency with community standards.
>     2. LLMs and agent frameworks: We selected widely-adopted models (Claude 3.7 Sonnet, GPT-4.1, DeepSeek-V3.1) and frameworks (SWE-agent, OpenHands, Aider) based on their demonstrated effectiveness in software engineering and cybersecurity tasks [2-6].
>     3. We are currently running new experiments with the newest models and scaffolds (e.g., Claude Sonnet 4.5). Our preliminary results show that these techniques can achieve superior performance than existing ones in the current version of paper (i.e., about 39% correct and secure rate with Claude Sonnet 4.5 and SWE-agent). We will update our results once we complete experiments of new experiments.
>
> References:
>
> [1] Zan, Daoguang, et al. "Multi-swe-bench: A multilingual benchmark for issue resolving." arXiv preprint arXiv:2504.02605 (2025).
>
> [2] Jimenez, Carlos E., et al. "SWE-bench: Can Language Models Resolve Real-world Github Issues?." ICLR. 2024.
>
> [3] Wang, Zhun, et al. "CyberGym: Evaluating AI Agents' Cybersecurity Capabilities with Real-World Vulnerabilities at Scale." arXiv preprint arXiv:2506.02548 (2025).
>
> [4] Lee, Hwiwon, et al. "SEC-bench: Automated Benchmarking of LLM Agents on Real-World Software Security Tasks." NeurlPS. 2025.
>
> [5] Udeshi, Meet, et al. "D-CIPHER: Dynamic Collaborative Intelligent Multi-Agent System with Planner and Heterogeneous Executors for Offensive Security." arXiv preprint arXiv:2502.10931 (2025).
>
> [6] Shao, Minghao, et al. "CRAKEN: Cybersecurity LLM Agent with Knowledge-Based Execution." arXiv preprint arXiv:2505.17107 (2025).

---

> ### Author Response · Authors · 2025-11-22
> **Official Response of Submission460 by Authors (2/N)**
>
> **Q3**
> > Unclear Language Coverage and Extensibility: The specific coverage of programming languages is not explicitly defined in the main text, though the prompt templates in Appendix H explicitly specify a "C/C++ code repository". If the dataset is exclusively C/C++, this should be stated more clearly as a scope limitation. Furthermore, it would be beneficial to discuss whether the proposed construction method can be easily extended to other languages beyond C/C++.
>
> **A3**
>
> 1. Language: We acknowledge the limitation in language coverage. In the revised version, we will explicitly state that SecureAgentBench is a C/C++ benchmark sourced from OSS-Fuzz, which primarily focuses on these two languages. This clarification will better position our work within the existing research landscape. Moreover, we emphasize the critical importance of C/C++ security: as memory-unsafe languages that underpin substantial cyber infrastructure, their security implications are particularly significant compared to memory-safe alternatives.
>
> 2. Extension: Although this paper focuses on C/C++ vulnerabilities, key components of our data construction pipeline are language-agnostic, including the two-stage vulnerability introduction discovery and PoC-based dynamic evaluation. Therefore, our methodology has strong potential for extension to other programming languages. We will discuss this promising direction for future work in the revised version.
>
>
> **Q4**
> > Ambiguity in Vulnerability Verification: While Section 2.2 explicitly mentions manual quality assurance for test cases and requirements, it is unclear if the specific Vulnerability/CWE classifications underwent human verification to ensure they accurately reflect the root cause of the issues derived from OSS-Fuzz.
>
> **A4**
>
> We clarify that our benchmark involves two vulnerability classification systems:
>
> 1. OSS-Fuzz types: These vulnerability types are specifically designed for OSS-Fuzz scenarios, and are labeled and verified by the open-source maintainers during the vulnerability disclosure process on OSS-Fuzz.
> 2. CWE types: To ensure consistency with prior benchmarking work and facilitate comparison, we map OSS-Fuzz types to CWE categories based on the mapping rules manually established by SecRepoBench [1]. This mapping is documented in Table 7 of our appendix.
>
> In our revised version, we will provide more explicit details about these classification systems and the mapping process to make them clearer and more transparent.
>
> References:
>
> [1] Dilgren, Connor, et al. "Secrepobench: Benchmarking llms for secure code generation in real-world repositories." arXiv preprint arXiv:2504.21205 (2025).

---

### Author Response · Authors · 2025-11-22
**Update from Authors (22 Nov)**

Dear Reviewers,


We would like to begin by expressing our sincere gratitude for your thoughtful and constructive feedback on our manuscript. Your insightful comments have been invaluable in identifying areas for improvement, and we are fully committed to addressing each point with careful attention to detail.


We want to assure you that we are actively working on the revisions and will complete them as promptly as possible. Specifically, we are prioritizing the following key areas you have highlighted: 1) failure mode analysis of models and agents; 2) further analysis of SAST results; 3) experiments under new settings with additional analysis; and 4) presentation improvements, including clarity of statements and formatting. Please note that these represent our overarching revision priorities, and we will address each specific point raised in your individual reviews with corresponding detailed responses.


We deeply appreciate your patience and understanding as we work diligently to address your concerns. Upon completing our revisions, we will promptly update both our results and manuscript, and provide you with a comprehensive revision report detailing all changes made.


Thank you very much for your time and consideration. Have a good day!


Best regards,

Authors of Submission #460

---

### Note · Authors · 2025-12-29

I have read and agree with the venue's withdrawal policy on behalf of myself and my co-authors.